# An enantioselective four-component reaction via assembling two reaction intermediates

Sifan Yu [1,3], Wenju Chang [2,3], Ruyu Hua[1,3], Xiaoting Jie[1], Mengchu Zhang[1], Wenxuan Zhao [2], Jinzhou Chen[1], Dan Zhang[1], Huang Qiu [1] ✉, Yong Liang [2] ✉ & Wenhao Hu [1] ✉

A reaction intermediate is a key molecular entity that has been used in explaining how starting materials converts into the final products in the reaction, and it is usually unstable, highly reactive, and short-lived. Extensive efforts have been devoted in identifying and characterizing such species via advanced physico-chemical analytical techniques. As an appealing alternative, trapping experiments are powerful tools in this field. This trapping strategy opens an opportunity to discover multicomponent reactions. In this work, we report various highly diastereoselective and enantioselective four-component reactions (containing alcohols, diazoesters, enamines/indoles and aldehydes) which involve the coupling of in situ generated intermediates (iminium and enol). The reaction conditions presented herein to produce over 100 examples of four-component reaction products proceed under mild reaction conditions and show high functional group tolerance to a broad range of substrates. Based on experimental and computational analyses, a plausible mechanism of this multicomponent reaction is proposed.

Reaction intermediates are molecular entities that form from starting materials and readily convert to final products. The identification of such intermediates is necessary towards understanding the mechanism of a reaction[1,2]. However, these intermediates are highly reactive, short-lived, and their concentration is often very low[3]. Therefore, studies on this entity have remained a tremendous challenge in regarding to its structure, reactivity, and synthetic applications[3–5]. Over the years, remarkable progress has been made in identifying and characterizing the structure of these intermediate species via advanced physico-chemical analytical techniques (e.g. time-resolved spectroscopy, mass spectrometry)[6]. Recently, the interception of the reaction intermediate with bench-stable chemicals represents an appealing alternative, not only providing the evidence for the existence of specific intermediates, but also offering an efficient approach to observe its chemical reactivity[4,7,8]. Additionally, this method opens an opportunity to discovering new chemical reactions, which could have a great impact on in the pharmaceutical, material and fine chemical industry[9].

Multicomponent reactions (MCRs) that involve multiple starting materials typically react in a stepwise manner to rapidly yield complex products in a greener and more economical manner due to having greater than three highly diversifiable starting materials[10–12]. Unlike traditional reactions, multicomponent reactions increase the accessible chemical space exponentially with each additional reaction component[13,14]. The discovery of MCRs, especially with a 'higher order' variant, are particularly challenging and have emerged as the frontiers in contemporary organic synthesis[15,16]. Furthermore, stereochemical identification can be challenging while the formation of multiple chemical bonds takes place while various types of highly reactive intermediates

[1]Guangdong Provincial Key Laboratory of Chiral Molecule and Drug Discovery, School of Pharmaceutical Sciences, Sun Yat-sen University, Guangzhou 510006, PR China. [2]State Key Laboratory of Coordination Chemistry, Jiangsu Key Laboratory of Advanced Organic Materials, Chemistry and Biomedicine Innovation Centre, School of Chemistry and Chemical Engineering, Nanjing University, Nanjing 210023, PR China. [3]These authors contributed equally: Sifan Yu, Wenju Chang, Ruyu Hua. ✉e-mail: qiuhuang@mail.sysu.edu.cn; yongliang@nju.edu.cn; huwh9@mail.sysu.edu.cn

are rapidly produced in the process[10]. For example, the Ugi four-component reaction utilizes an aldehyde, amine, carboxylic acid, and isocyanide to afford α-aminoacyl amide derivatives is the most well-known multicomponent reaction and was first reported by Ugi and coworkers in 1959[11]. An enantioselective version of this reaction has only recently been published by Tan and coworkers in 2018[17], although its four-component product owns only one chiral center. Considering increasing demand for the development of highly attainable and selective MCRs, the robust and general strategy that can introduce new transformations with a 'higher order' variant is highly needed[15,16].

As shown in Fig. 1a, previously developed three-component reactions involved using a bench-stable chemical to trap the reaction intermediate generated from traditional reactions[18]. Through the sequential assembly of reaction intermediates and trapping reagents, rare examples of MCRs with four variants (four-component reactions) have been discovered in history[17,19,20]. Given that various bench-stable components could successfully trap the transient reaction intermediate, we further questioned whether it would be possible to trap such intermediate with another intermediate generated by another independent process. The resultant process would be used to discover MCRs with four variants if distinct substrates generate both reaction intermediates[15,21,22]. However, this strategy suffered numerous challenges in a spatial-temporal version, requiring each intermediate should be generated simultaneously at similar rates. The reaction of these two intermediates occurred in a highly efficient, specific, and selective manner[3,8,23]. Otherwise, incompatibility of various reactants, uncontrollable sequence of chemical bond formation, and irreversible side reactions would hamper the development of such types of transformations[24]. Our previous findings suggested that the catalyst system heavily influences the generation of reaction intermediates and their compatibility with trapping reagents (Fig. 1b, c)[25–30]. With an proper choice of catalytic systems[31–45], simultaneous generation of two types of reaction intermediates with compatible chemical reactivity via distinct catalytic cycles could be possible[3,23], thus providing an opportunity to be of use in discovering different MCRs.

It has been extensively explored that diazoester could react with alcohol under the catalysis of transition metal to afford oxonium ylide intermediate (or its enol form), a highly reactive nucleophilic species, which various electrophiles could trap according to previous findings[18]. In addition, the generation of α,β-unsaturated iminium intermediate using indole and aldehyde under the catalysis of both Brønsted acid and amine via Knoevenagel-type reaction was well-documented[46]. Coupling the two intermediates above would finally deliver the desired 4CRs (Fig. 1d). To our best knowledge, this intermediate cross interception above has not been disclosed. Although this designed strategy was straightforward, numerous potential side reactions would irreversibly consume newly generated reactive intermediates to produce an array of undesired side products[47–49]. For example, the metal carbene species has been reported to react with various reagents[50,51], including aldehydes, imines, indoles, and amines, in the chemical system, generating those undesired ylides or zwitterionic intermediates[18]. Other potential side reactions of oxonium ylide with aldehydes or imines are also possible in this design[30]. Furthermore, the stereocontrol issue remains a formidable challenge, especially involving two vicinal quaternary/tertiary stereocenters[52].

In this work, we report various highly diastereoselective and enantioselective four-component reactions (containing alcohols, diazoesters, enamines/indoles and aldehydes) which involve the coupling of in situ generated intermediates (iminium and enol).

## Results and discussion
### Substrate scope
To validate our conceptual hypothesis, we firstly explored the proposed four-component reaction with 4-bromobenzyl alcohol **1a**, tert-butyl phenyldiazoesters **2a**, 2-methylindole **3A**, and 3-phenylpropiolaldehyde

**4a** as model substrates. Thorough optimizations involved the evaluation of reaction parameters and ultimately led to the following optimal reaction conditions: 5 mol% Pd(CH$_3$CN)$_2$Cl$_2$, 10 mol% SPINOL-derived chiral phosphoric acid **6a**, and 50 mol% 2,5-bis(trifluoromethyl)aniline **5a** at −20 °C in DCE (Supplementary Tables 1 and 3).

Under the optimized conditions, the substrate scopes of this asymmetric four-component reaction (A4CR) were then investigated (Fig. 2, Supplementary Fig. 1, Supplementary Data). Initially, we examined the primary alcohols **1** and found that benzyl alcohols with diverse functional groups were proven to be suitable substrates (**7–10**). We were pleased to observe that heteroaromatic alcohols, such as those with furanyl and benzothienyl groups, could readily be employed efficiently (**11–12**). Moreover, cinnamyl alcohol, allyl alcohol, and 3-phenyl-2-propyn-1-ol were highly applicable to the present reaction and gave desired products in synthetically useful yields without compromising the diastereo- and enantioselectivity (**13–15**). Concerning the α-diazoesters, a variety of *ortho*-, *meta*-, and *para*- substituted α-aryl-α-diazoesters with various functional groups underwent efficient and highly A4CRs (**16–21**), and the absolute configuration of **21** was confirmed by X-ray crystallographic analysis (Supplementary Fig. 5). 2-Ethyl indole also resulted in high ee (93%) of the desired product. Notably, 2,5 or 2,6-disubstituted indoles containing bromo, alkyl, phenyl, Bpin, amino, and sulfanyl groups were converted to their corresponding products in satisfying yields with good to excellent enantioselectivity (**22–29**). Concerning the aldehyde, various substituted (hetero)aromatic ynals were well tolerated in condition A, leading to corresponding products in good yield and diastereo- and enantioselectivity (**30–37**). Additionally, we investigated the substrate limitation of this A4CR. For example, secondary alcohol, tertiary alcohol, unprotected indoles, indole with a phenyl group at 2-position, and simple aryl aldehyde could not convert to their related products under the catalytic condition A (Supplementary Fig. 1b).

We next explored the extension of the present A4CR to other compounds containing enamine motifs. 4-(1-Phenylvinyl)morpholine was identified as a proper substrate under condition B (Supplementary Tables 2 and 4). Figure 3 summarizes the detailed results for this A4CR. A broad spectrum of alcohols, α-diazoesters, enamines, and aldehydes was then investigated (**38–86**) (Supplementary Fig. 2, Supplementary Data). Like the A4CRs above, the primary alcohols and α-diazoesters with various functional groups and heterocycles were well-tolerated, delivering the corresponding four-component products with satisfied reactivity and selectivity (**38–65**). The absolute stereochemistry of **41** was concluded by X-ray crystallographic analysis of its derivative (Supplementary Fig. 6). Of particularly noted, we were delighted to find that α-alkyl-α-diazoesters also readily participate in this reaction (**66** & **67**), demonstrating the diversity of substituents. Enamines with different substituents were also amenable to this protocol, affording the desired products (**68–70**) in good yields and with excellent diastereo- and enantioselectivities. We next evaluated the scope of aldehydes. Diverse propiolaldehydes showed excellent reactivity regardless of whether the substituents were electron-donating or electron-withdrawing groups (**71–77**). To further broaden the utility of this protocol, we further explored α,β-unsaturated aldehydes. Interestingly, α,β-unsaturated aldehydes bearing ester and ketone groups were perfectly tolerated under the condition B, furnishing the desired products (**78–82**) in good yields and with excellent diastereo- and enantioselectivities. Moreover, aromatic aldehydes could also function efficiently in this four-component coupling protocol. For example, *para*-nitro aldehyde and its 3-methyl analog readily reacted with various alcohols in the current protocol (**83–86**) (Fig. 3, Supplementary Fig. 2a). Additionally, we investigated the substrate limitation of this A4CR. For example, secondary alcohol, tertiary alcohol, diazoacetate, alkyl enamine and simple aryl aldehyde could not convert to their related products under the catalytic condition B (Supplementary Fig. 2b).

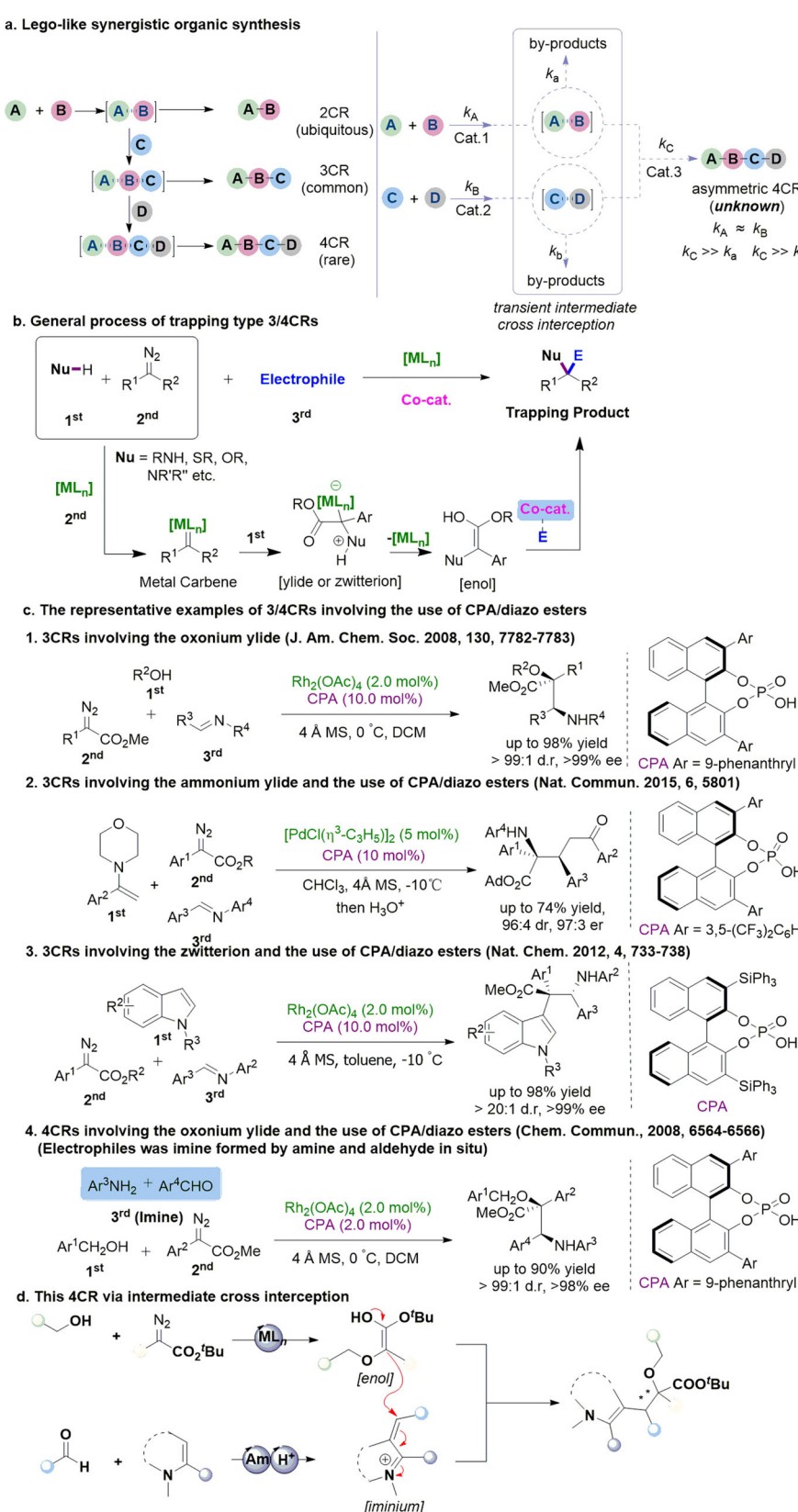

**Fig. 1 | Patterns of chemical reactions. a** Lego-like synergistic organic synthesis. **b** General process of trapping type MCRs developed by our group. **c** The representative examples of 3/4CRs involving the use of CPA/diazo esters. **d** This 4CR via intermediate cross interception. NOTE: Nu is short for nucleophile, Co-cat. Is short for co-catalyst, CPA is short for chiral phosphoric acid, [MLn] is short for metal catalyst, [Am] is short for amine catalyst, [H+] is short for Brønsted acid catalyst.

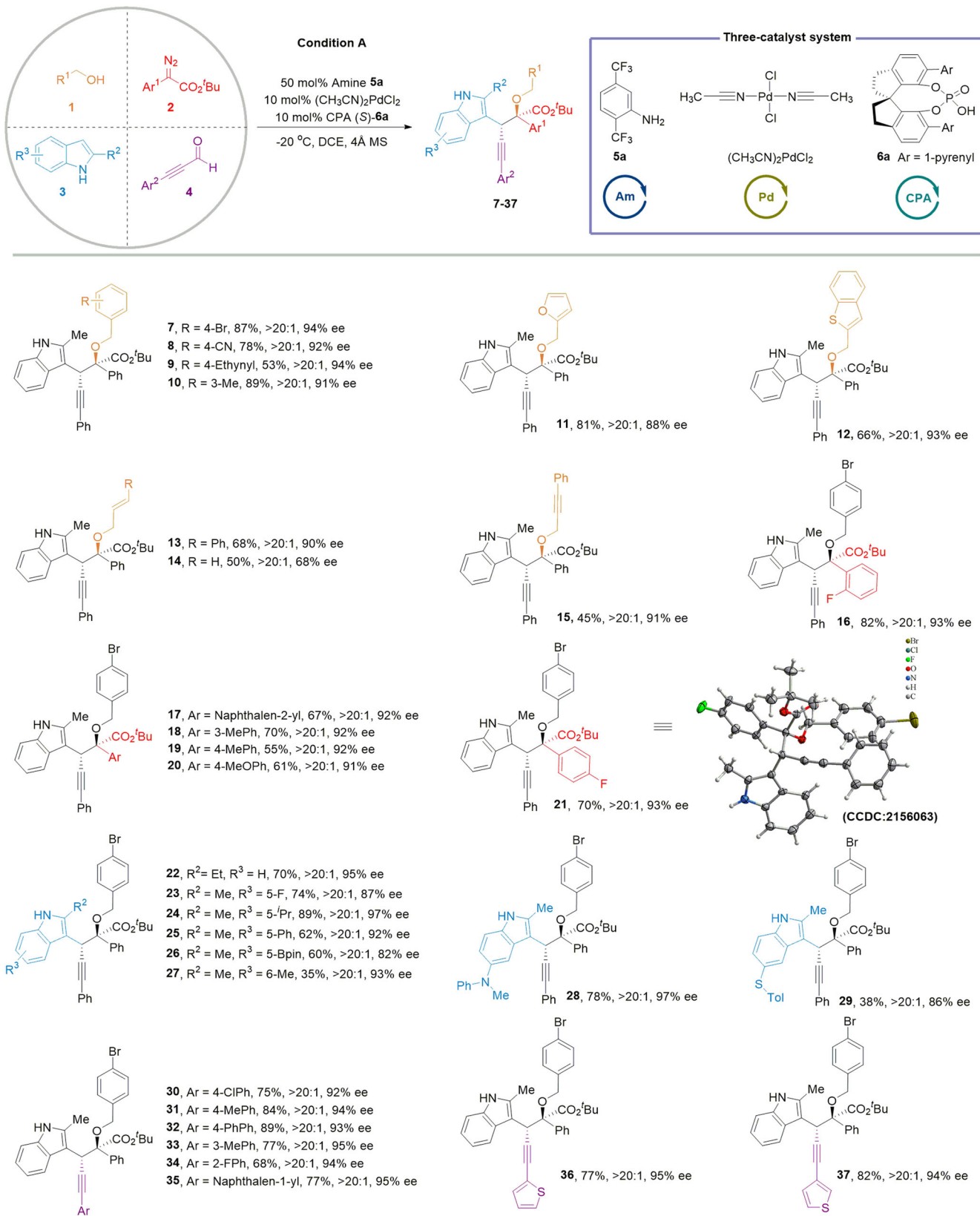

**Fig. 2 | Substrate scope of enantioselective four-component reactions of alcohol, diazoester, indole and aldehyde derivatives.** Standard conditions: **1/2/3/4/5a/[Pd]/6a** = 0.6/0.3/0.2/0.6/0.1/0.01/0.02 mmol. **2, 3** in 4.0 mL dry DCE was added into a solution of **1, 4, 5a** (50 mol%), [Pd] (5 mol%), **6a** (10 mol%), and 4 Å MS (50 mg) in 4.0 mL dry DCE via a syringe pump for 60 min, and the resulting mixture was stirred for another 12 h at −20 °C.

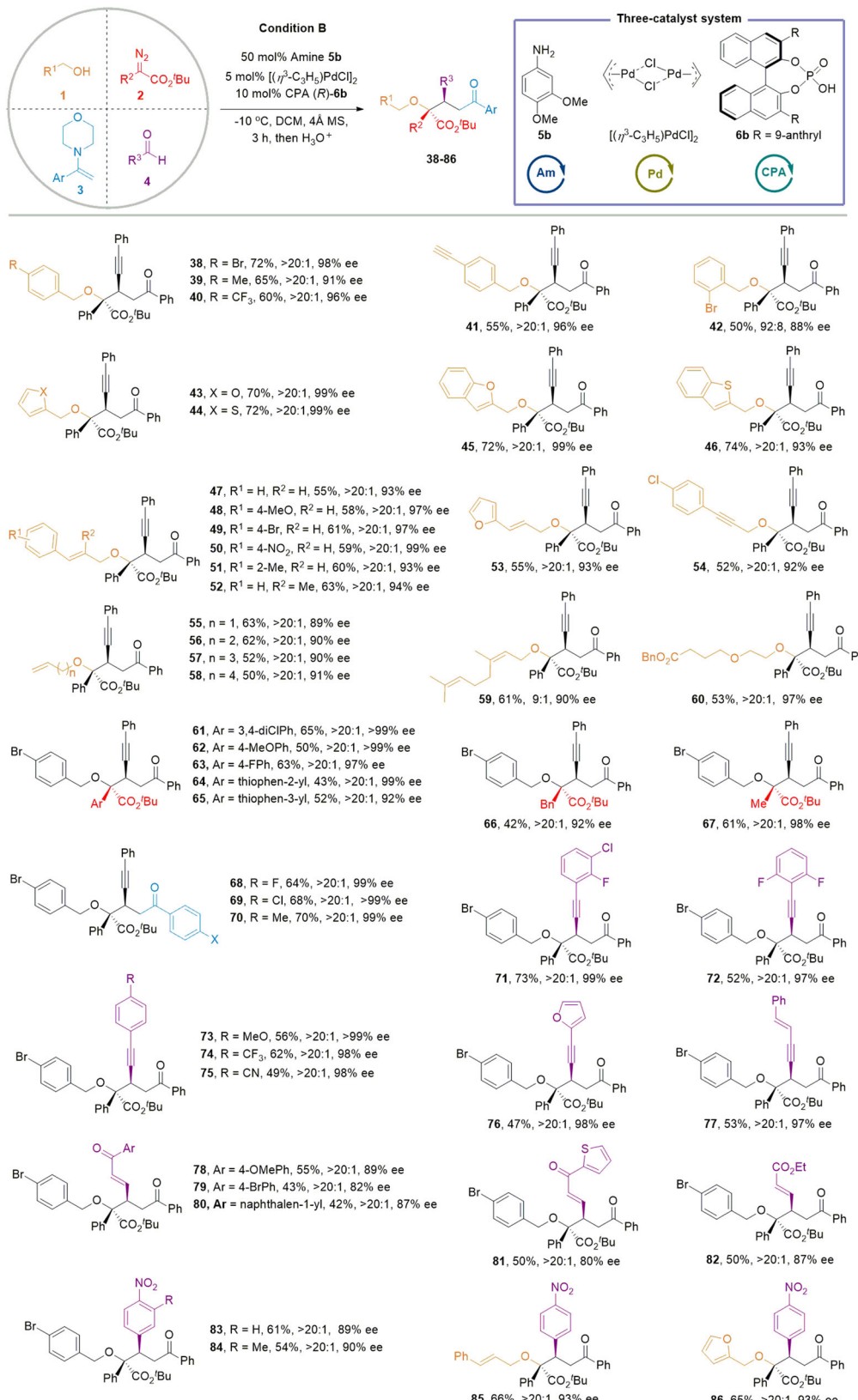

**Fig. 3 | Substrate scope of enantioselective four-component reactions of alcohol, diazoester, enamine and aldehyde derivatives.** Standard condition: **1**/**2**/**3**/**4**/**5b**/[PdCl(allyl)]$_2$/**6b** = 0.18/0.18/0.18/0.15/0.075/0.0075/0.015 mmol, **2** and **3a** in 1.0 mL dry DCM are added into a solution of **1**, **4**, [PdCl(allyl)]$_2$, **6b**, aniline **5b** and 100 mg 4 Å MS in 1.5 mL dry DCM via a syringe pump under a nitrogen atmosphere for 3 h, and the resulting mixture was stirred for another 1 h at −10 °C.

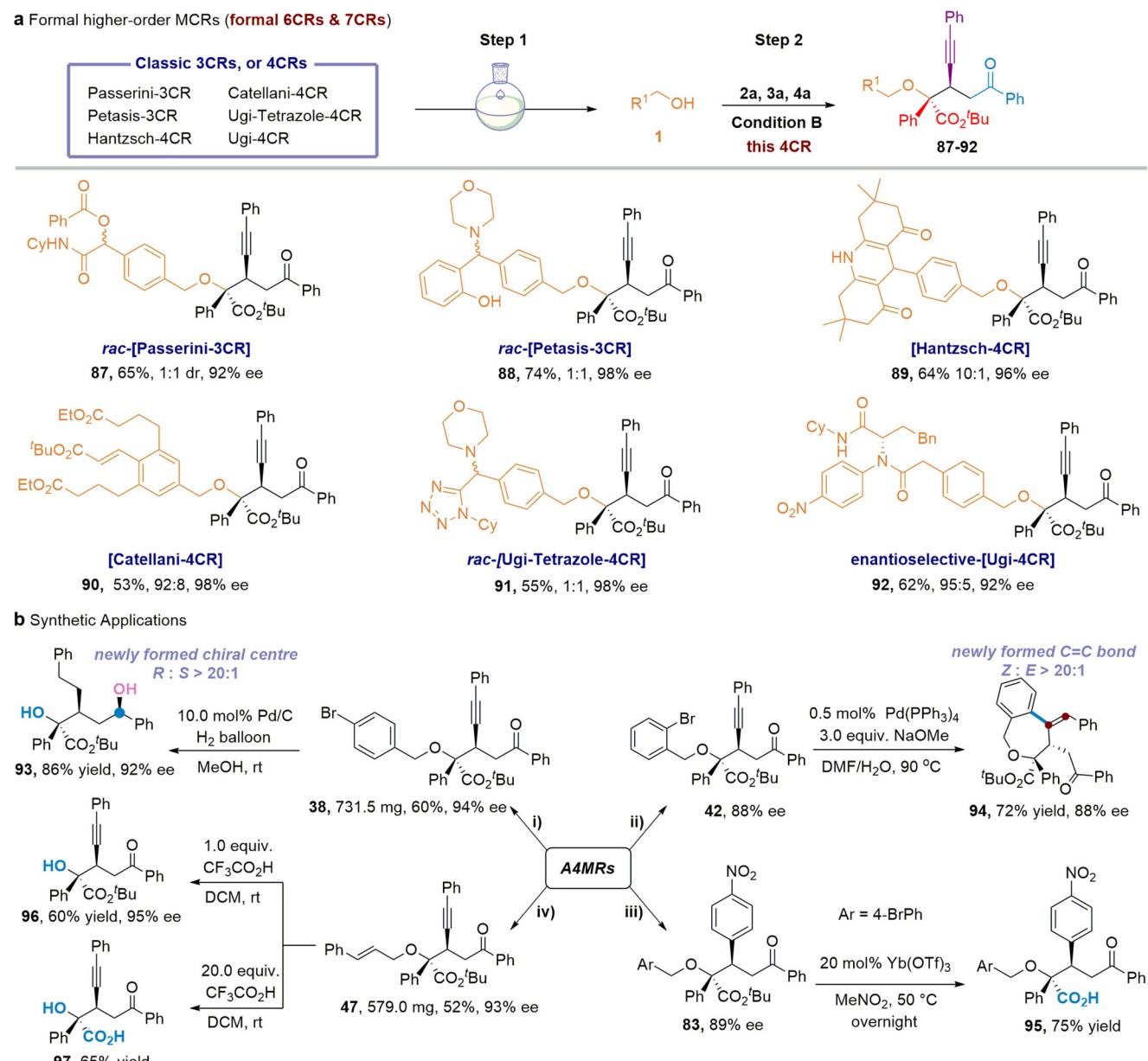

**Fig. 4 | Formal higher-order MCRs design and synthetic applications. a** Formal higher-order MCRs design. **b** Synthetic applications. NOTE: The other diastereomer **87'** of compound **87** has the same ee value (92% ee); The other diastereomer **88'** of compound **88** has the same ee value (98% ee); The other diastereomer **91'** of compound **91** has the same ee value (98% ee).

## Formal higher-order MCRs design and synthetic applications

The high functional group tolerance encouraged us to further investigate the practical utility of this protocol. Notably, several classic MCR products are also compatible with this enantioselective 4CR, including Passerini-3CR[11], Petasis-3CR[53], Hantzsch-4CR[54], Catellani-4CR[55], Ugi-Tetrazole-4CR[14], and enantioselective Ugi-4CR[11,17], affording the desired products (**87–92**) with 6 or 7 formal variants in satisfying yields and excellent enantioselectivities in two steps (Fig. 4a, Supplementary Fig. 3, Supplementary Methods). We also performed several transformations of the obtained four-component products, shown in Fig. 4b. For example, when **38** was reduced by H₂ in the presence of Pd/C, the product **39** containing three chiral carbon centers was obtained in satisfying yield with excellent diastereoselectivity, and its absolute configuration was predicted by Electronic circular dichroism (Supplementary Fig. 7). **42** could readily be converted into benzoxepine **94** in 72% yield via a palladium-mediated reductive Mizoroki–Heck cyclization, and we were delighted to observe the excellent diastereoselectivity and geometric selectivity (Z:E > 20:1) of the desired product. Notably, treatment of **83** with 20 mol% Yb(OTf)₃ could effectively remove the *tert*-butyl group to form the carboxylic acid **95** in 75% yield without losing the diastereo- and enantioselectivity. In addition, treatment of **47** with 1.0 equiv. trifluoroacetic acid (TFA) readily afforded the formal product of the four-component reaction with H₂O (**96**) in 60% yield. When 20.0 equiv. TFA was used, and we obtained the α-hydroxyl acid **97** in 65% yield without losing the diastereo- and enantioselectivity (Supplementary Methods, Supplementary Fig. 4).

## Mechanistic investigations

DFT calculations were performed to explore the mechanism of this A4CR and the origin of stereoselectivity (Supplementary Discussion, Supplementary Figs. 8–13). Combining with the mechanistic studies (Supplementary Discussion), a catalytic cycle was proposed in Fig. 5a. In the amine-CPA catalyzed cycle, intermediate **INT-2** formed via a CPA

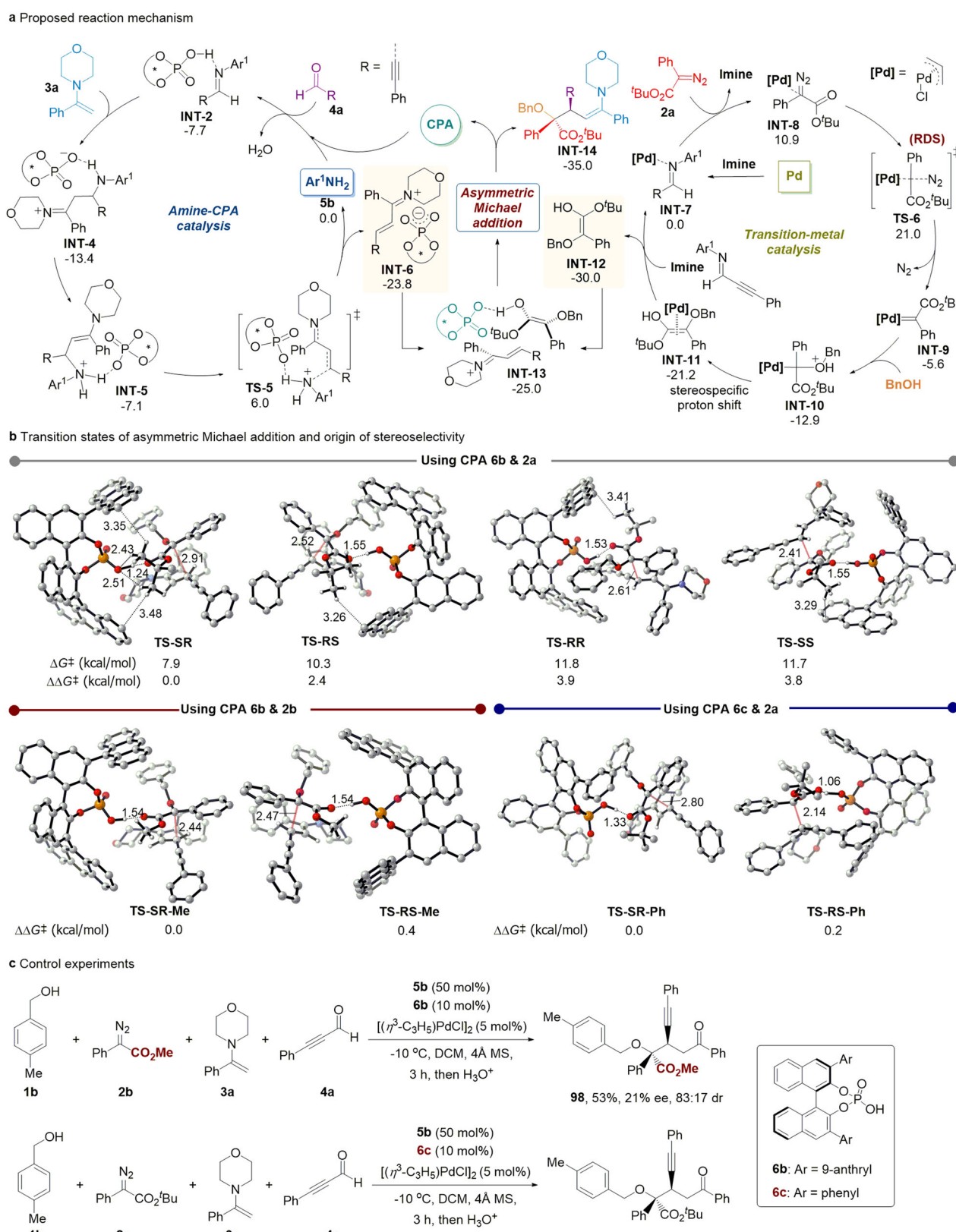

**Fig. 5 | Mechanistic studies of enantioselective four-component reaction.**
**a** Proposed catalytic cycle. **b** Optimized transition state geometries of symmetric
Michael addition and origin of stereoselectivity. **c** Control experiments. In **a** and

**b**, the Gibbs free energies were computed with CPCM(DCM)-M06-2X/6-311+G(d,p)
[SDD for Pd]//CPCM(DCM)-B3LYP/6-31G(d)[LANL2DZ for Pd]. Most hydrogen
atoms are omitted for clarity. All distances are in angstroms.

catalyzed nucleophilic addition of aniline **5b** on aldehyde **4a** and elimination of a water molecule. Subsequently, **INT-2** was attacked by enamine **3a**, resulting in the formation of **INT-4**. After a subsequent CPA anion-assisted proton transfer, **INT-5** was generated, which delivered the key intermediate **INT-6** and aniline **5b** via transition state **TS-5**. Calculations showed that the overall energy barrier was 19.4 kcal/mol for the formation of intermediate **INT-6**. In the transition-metal-catalyzed cycle, the dissociation of palladium dimer precatalyst by imine gave **INT-7**, which took an exchange with diazoester **2a** to give **INT-8**. Then palladium-catalyzed decomposition of diazoester **2a** led to carbene intermediate **INT-9** and the nitrogen release. Oxonium ylide **INT-10** was formed from carbene **INT-9** by the attack of benzyl alcohol and then transformed into a more stable Pd-associated enol **INT-11** through a quick intramolecular [1,3]-proton shift. In an exergonic process, an exchange between **INT-11** and imine gave free enol **INT-12** and regenerated palladium catalyst. For the formation of free enol **INT-12**, the overall barrier was 21.0 kcal/mol. Finally, cross interception of the two active intermediates, **INT-6** and **INT-12**, took place in the chiral pocket of CPA **6b** anion, providing the four-component product with simultaneously regenerated the CPA **6b**. The main challenge of cross interception of the in situ formed two active intermediates is that they need a comparable formation rate to avoid consumption before cross interception occurs. DFT calculations demonstrated that the generation of the two active intermediates, **INT-6** and free enol **INT-12**, had matched energy barriers (19.4 kcal/mol versus 21.0 kcal/mol), enabling the four-component reaction to proceed smoothly.

Asymmetric Michael addition afforded products with successive $\alpha,\beta$-stereocenters, and the stereoisomeric TSs are presented in Fig. 5b. Among them, **TS-SR** had lower free energy than that of **TS-RS** and **TS-SS** by 2.4 and 3.8 kcal/mol, respectively. This is in accordance with the 92% ee and 94:6 dr obtained experimentally. In comparing **TS-SR** with the others (**TS-RS**, **TS-RR**, and **TS-SS**), favorable C-H…π and C-H…O interactions between CPA and the large *tert*-butyl group in enol led to the preference for the formation of the predominant product. While less C-H…π interactions were present in the other transition states. This result indicated that non-covalent interaction plays a critical role in controlling stereoselectivity[56]. Additional DFT calculations were conducted to clarify the importance of nonbonding interactions between CPA and substrate (Fig. 5b, bottom). Replacing *tert*-butyl group by a much smaller methyl group in enol, the C-H…π and C-H…O interactions disappeared in the asymmetric Michael addition transition states. As a result, the relative free Gibbs energy difference between **TS-SR-Me** and **TS-RS-Me** ($\Delta\Delta G^{\ddagger} = 0.4$ kcal/mol) became smaller than that between **TS-SR** and **TS-RS** ($\Delta\Delta G^{\ddagger} = 2.4$ kcal/mol), indicating a poor stereoselectivity without a *tert*-butyl group. Moreover, using **6c**, a CPA with a smaller chiral pocket, and diazoester **2a** with a large *tert*-butyl group, **TS-SR-Ph** and **TS-RS-Ph** became flexible where these non-covalent interactions were also not present. The significant decrease in relative free Gibbs energy ($\Delta\Delta G^{\ddagger} = 0.2$ kcal/mol) demonstrated that the 9-anthryl group was essential in determining the stereoselectivity. Control experiments were carried out and shown in Fig. 5c, and poor selectivity was obtained using either CPA **6b** and diazoester **2b** (21% ee, 83:17 dr) or CPA **6c** and diazoester **2a** (6% ee, 84:16 dr). It is clear that both the *tert*-butyl group in diazoester (**2a**) and the 9-anthryl group in CPA (**6b**) are critical to enabling the asymmetric Michael addition with high selectivities through non-covalent interactions.

In conclusion, the four-component reactions of alcohols, diazoesters, enamines/indoles, and aldehydes are developed, affording the corresponding products in satisfying yields with good to excellent chemo-, diastereo- and enantioselectivity. These four-component reactions proceed under mild reaction conditions and show high functional group tolerance. In addition, some classic MCRs products are well-compatible with the present four-component processes, affording several unprecedented examples of formal higher-order MCRs. Mechanism studies demonstrated that the comparable energy barriers of iminium and enol intermediates formation through independent processes ensure the success of this enantioselective four-component reaction. Both CPA and diazoester are critical to our reaction and responsible for acquiring excellent stereoselectivity. Furthermore, these multicomponent methods open more opportunities for developing the MCRs by assembling two reaction intermediates generated by independent catalytic processes.

## Methods
### General procedure for enantioselective four-component reactions of alcohol, diazoester, indole and aldehyde derivatives
To a flame-dried 20-mL Schlenk flask charged with a magnetic stirring bar, alcohol **1** (0.6 mmol), aldehyde **4** (0.6 mmol), 2,5-bis(trifluoromethyl)aniline **5a** (50 mol%), Pd (CH$_3$CN)$_2$Cl$_2$ (10 mol%), (*S*)-**6a** (10 mol%) and 4 Å MS (100 mg) in DCE (4.0 mL), was added a mixture of diazoacetate **2** (0.3 mmol) and indole **3A-3I** (0.2 mmol) in DCE (4.0 mL) for 1.0 h via a syringe pump at −20 °C. The mixture was stirred for another 12 h at −20 °C. After the completion of the reaction, the reaction mixture was filtrated and the filtrate was evaporated in vacuo to give the crude product. And then the crude product was purified by flash chromatography on silica gel (eluent: PE: EA = 20: 1) to afford the pure products.

### General procedure for enantioselective four-component reactions of alcohol, diazoester, enamine and aldehyde derivatives
To a flame-dried 10-mL Schlenk flask charged with a magnetic stirring bar, 3,4-dimethoxyaniline **5b** (0.075 mmol), 5.0 mol% [PdCl(allyl)]$_2$, 10.0 mol% (*R*)-**6b**, alcohol **1**(0.18 mmol), aldehyde **4** (0.15 mmol) and 100 mg 4 Å MS in 2 mL DCM were sequentially added at −10 °C. Diazoacetate **2** (0.18 mmol) and enamine **3a–3c** (0.18 mmol) dissolved in DCM (1.5 mL) were added by syringe pump over 3 h. The mixture was stirred for another 1.0 h at −10 °C. After the completion of the reaction, the reaction mixture was filtrated and the filtrate was evaporated in vacuo to give the crude product. And then the crude product was purified by flash chromatography on silica gel (eluent: PE/EA = 50:1 ~ 30:1, v/v) to give the pure product (If the reaction was carried out on more than 0.15 mmol scale, the decreased ee value of corresponding product might be detected).

## Data availability
CCDC (2156063&1948222) contains the supplementary crystallographic data for this paper. These data can be obtained free of charge from the Cambridge Crystallographic Data Centre via www.ccdc.cam.ac.uk/data_request/cif. Experimental procedures, characterization of all new compounds, computational details, Supplementary Tables and Supplementary Figures are available in the Supplementary Information. The authors declare that the data supporting the manuscript are included in the manuscript and supplementary materials.

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

## Acknowledgements

We thank Michael. P. Doyle for comments and suggestions on the preparation of the manuscript. We thank Bin Tan for generously providing chiral SPINOL phosphoric acid **S6l**. We also thank Chi Xiao, Gengxin Liu, Shanshui Meng, Zhenhui Kang, Zongwen Mo for giving help and suggestions in completing the project. W.H. and H.Q. acknowledge the financial support from Guangdong Innovative and Entrepreneurial Research Team Program (No. 2016ZT06Y337). W.H. and H.Q. acknowledge the National Natural Science Foundation of China (NO. 92056201). H.Q. acknowledges support from NSFC grant 21801255. Y.L. acknowledges support from the Fundamental Research Funds for the Central Universities (020514380253, 020514380277), the Natural Science Foundation of Jiangsu Province (BK20211555), and the Jiangsu Innovation & Entrepreneurship Talents Plan. We thank the High Performance Computing Center (HPCC) of Nanjing University for doing the numerical calculations in this paper on its blade cluster system.

## Author contributions

S.Y., H.Q. and W.H conceived and designed the project; S.Y. and R.H. developed the catalytic asymmetric four-component reactions, and conducted most of the experiments; M.Z., X.J. performed parts of substrate scope experiments and synthetic applications; W.C. and W.Z. conducted the DFT calculations and provided mechanism analysis; J.C. conducted the ECD calculations; Y.L. directed the DFT calculations and mechanism; D.Z. helped the direction of the project; S.Y., W.C., R.H., H.Q., Y.L., and W.H. prepared the manuscript.

## Competing interests

The authors declare no competing interests.
