## [Peer Review File · Nature Communications]

REVIEWER COMMENTS

Reviewer #1 (Remarks to the Author):

Multicomponent reactions are extremely important in organic synthesis because they provide a complex structural unit by stitching readily available molecules together. Certainly, there has been a great deal of success in achieving such a reaction in the literature. However, the enantioselective version of those process are not so common especially when it comes to four component reactions. As mentioned by the authors the first successful example includes the enantioselective Ugi-4CR utilizing an aldehyde, an amine, a carboxylic acid and an isocyanide. This is followed by the reports from Feng and Hoveyda (Ref 21 and 22). In this report, the authors reported the first example enantioselective of four-component reaction of alcohols, diazoesters, enamines/indoles and aldehydes. The reaction affords products in good yields with good to excellent chemo-, diastereo- and enantioselectivity. The reaction seems to be very general and this referee has no doubt about its potential in terms of creating chiral building block and developing analogues multicomponent reactions of high magnitude. This referee is in favour supporting this work for publication in this journal after addressing the following comments.

1) The reports lack a variety of reviews on multi-catalysis where more than one catalysts work in tandem in relay or cooperative fashions. Just to quote a few: *Tetrahedron*, 2013, 69, 7171; *OBC*, 2012, 10, 211, *Chem. Rev.*, 2005, 105, 1001; *Coord. Chem. Rev.*, 2004, 248, 2365; *OBC*, 2015, 13, 8116; *Chem. Asian J.* 2010, 5, 422, *ACS Catal.* 2021, 11, 3891 etc.

2) The control experiments are missing: 1) the reaction of enol (or its keto equivalent) with aldehyde and enamines/indoles 2) the reaction of alcohol and diazo compound with enamines/indoles equivalent (for example, appropriately substituted indole-3-carbinol)

Reviewer #2 (Remarks to the Author):

This manuscript by Yu and coworkers reports an efficient enantioselective four-component reaction via in-situ two intermediates. The reaction proceeds with four-component that involve alcohols, diazoesters, enamines/indoles, and aldehydes and furnishes the corresponding coupling products in good yield under mild reaction conditions and shows broad substrate scope. Besides evaluating the synthetic scope of the reaction and the synthetic applications, they also elucidate its mechanism involving DFT computational work that was conducted to support the stereoselectivity of the Michael addition. However, the computations currently of the transition-metal catalysis are discussed in such a way as to suggest that these calculations support the current experimental

results, which is currently not the case. Therefore, I either recommend the authors do the additional calculations in order to support the experimental data (such as a larger database method) or rewrite the discussion of the current DFT calculations. Specific comments and questions are provided below:

1. Though the authors propose the imine was involved in the Pd catalysis to close the catalysis cycle, the current calculations suggest that INT-11 to INT-7 (Fig.5a) is unlikely to happen smoothly, as the intermediates' barrier of 21.2 kcal/mol will not result in good yield at -10 oC. Additional calculations on the concerted pathway are needed to support the current experimental evidence.

2. The authors mention "the substrate scope and limitation of this four-component reaction were then investigated (Fig.2)." on page 3, line 99, but there is no limitation result in Fig.2 or others. Please include some limitation results in the manuscript, it will be better for the chemists to study the research. For example, secondary alcohols, protected indoles, phenyl group instead of the methyl group at 2- position of indole, or N-heteroaryl group, etc.

3. The authors should write the conditions in the footnote of Fig.2 and 3, such as the mmol of each substrate and the syringe addition time, etc.

4. In Fig.2, the structure of products 22 to 27, the authors should use R2 instead of R1 and use R3 instead of R2 to be consistent with the general structure of substrate 3 on the top of Fig.2.

5. It's better to involve the TS barrier in the proposed reaction mechanism (Fig.5a).

6. I would like to suggest the authors use a larger database method to calculate the stereoselectivity including a Boltzmann calculation in further research.

Reviewer #3 (Remarks to the Author):

1.) The noteworthy results of this paper is an example of a highly enantioselective four-component reaction with a large reaction scope.

2.) Established literature for enantioselective reactions with diazaesters do exist, but are typically only two-three component reactions. This being a four component reaction with high ee and dr as well as a DFT-mechanistic study and large scope add to its value. Additionally, having a racemic product to compare to the enantioselective product further validated these claims. I would recommend to have a figure introducing to the readers previous work of enantioselective three/four component reactions involving the use of CPA/diaza esters, see: *Molecules* 2021, 26, 6563

3.) Based on chiral HPLC/NMR and previous work involving the use of these catalysts, and DFT calculations, the results of this work are sufficiently supported.

Analysis of the %ee via the chiral HPLC chromatogram needs to be relooked at. When comparing the racemic to the chiral reaction, the amount measured for the minor peak is consistently smaller when compared to the racemic one. Therefore a higher %ee is being measured than what is actually obtained. The chiral HPLC chromatogram needs to be zoomed in further and the area measured needs to account for the entire curve. Additionally, the grammar/sentence structure needs to be revised for the whole paper. Such recommendations were given up to line 49.

The methodology is sound and meets the expected standards.

Given the large reaction scope, optimization table, and previous literature, it is expected that this work would be reproducible.

Reviewer,

Kevin Schofield

Review Comments

Title

An enantioselective four-component reaction via assembly of two reaction intermediates

Abstract

Line 6: A Reaction intermediate... how ~~the~~

Line 7: starting materials convert into ~~the~~ final products in a reaction. An intermediate is usually unstable, highly reactive, and short-lived.

Line 10: Trapping experiments ~~with additional chemicals~~ are an efficient and...

Line 11-13: ~~More importantly,~~ This trapping strategy opens an opportunity to discover new multicomponent reactions (MCRs). ~~and theoretically, novel multicomponent reactions with a 'higher order' variant could be developed via assembling two reaction intermediates.~~

Line 14: Herein, we report various highly diastereoselective...

Line 15-18 ... which involve the coupling of *in situ* generated ~~two~~ intermediates (iminium and enol). The reaction conditions presented herein to produce over 100 examples of four-component reaction products proceed under mild reaction conditions and show high functional group tolerance to a broad range of substrates.

Line 19: Notably, the present...

Line 20-21: Based on experimental and computational analyses, a plausible mechanism of this multicomponent reaction is proposed.

Introduction + Results.

Line 25-28: Reaction intermediates are molecular entities that form from starting materials and readily convert to final products. The identification of such intermediates is necessary towards understanding the mechanism of a reaction. However, these intermediates are highly reactive, short-lived, and their concentration is often extremely low.³

Line 30: Remarkable progress has been ~~achieved~~ made

Line 37: in the pharmaceutical...

Line 38-41: Multicomponent reactions (MCRs) that involve multiple starting materials typically react in a stepwise manner to rapidly yield complex products in a greener and more economical manner due to having greater than three highly diversifiable starting materials.

Note from reviewer: Multicomponent reactions don't usually simultaneously react, they are typically stepwise in their reactivity. Try to avoid using such strong statements such as always.

Line 41-45: ~~Among these advances~~ Unlike traditional reactions, multicomponent reactions increase the accessible chemical space exponentially with each additional reaction component.

Line 47-49: Furthermore, stereochemical identification can be challenging while the formation of multiple chemical bonds takes place while various types of highly reactive intermediates are rapidly produced in the process.

Line 49: For example, the Ugi four-component reaction utilizes an aldehyde, amine, carboxylic acid, and isocyanide to afford α -aminoacyl amide derivatives is the most well-known multicomponent reaction and was first reported by Ugi and coworkers in 1959.¹³ An enantioselective version of this reaction has only recently been published by Tan and coworkers in 2018

The rest of the paper needs to be rewritten as every sentence contains grammatical mistakes.

Figure 1, b: The arrow going from the enol to the iminium is not correct. During nucleophilic attack onto an electrophile, you draw the arrow going from the bond to an atom. Draw the full arrow cascade, arrow from OH going into bond to form ester, then arrow from the double bond to atom of iminium, then arrows are needed on iminium to show flow of electrons. You should denote what Am and MLn stands for. It may be obvious for some, but not others.

Figure S1. Yields should be reported as 42 not 42.0, don't report yields with a decimal point.

Figure 2, **12**: 93% ee, not 94%.

15: 91% ee, not 92%.

18: 92% ee

21: Crystal structure should have higher quality, when zooming in it is really blurry.

22: 95% ee, not 96%.

23: 87% ee, not 88%.

24: Yield states 89% in SI, but 85% in paper, which is it?

25: 92% ee, not 93%.

27: 93% ee, not 94%.

28: 97% ee. Not 98% ee. Also, the Chiral IA graph could be zoomed in a bit. It looks like the area for the peak at 28 minutes is not fully covered.

29: 86% ee, not 87%.

34: SI states 68% yield, but paper states 77% yield, which is it?

36: 95% ee, not 96% ee.

37: 94% ee, not 95% ee.

39: 91% ee, not 92% ee.

40: 96% ee, not 98% ee.

43-45: Chiral IA graph should be zoomed in more; it might not make a difference but markers for minor product don't look they cover the full peak area. If

46: 93% ee, not 94%.

48: 97% ee, not 98%.

49: 97% ee, not 98%. Also, area under curve is not measured correctly for minor peak, need to zoom in more and measure correct area.

51: 93% ee, not 94%.

53: SI states 55% yield and paper states 60% yield, which is it. Paper states 93% ee, SI states 96% ee. Based on graph, 93% ee

54: 92% ee, not 93% ee.

55: 89% ee, not 90% ee.

58: 91% ee. not 90%.

59: Can just write 9:1, consistent with how you wrote it for previous molecules (20:1).

65: 92% ee, not 90%.

68: for ee so low the graph should be zoom in a little bit more.

72: Based on racemic peak, peak looks like it is supposed to be very wide and the peak area that you highlighted does not match how it would match in racemic peak. Again, this should be zoomed in greater so that the correct area can be visualized better.

73: Same issue as above.

74: Same issue as above, doesn't look like peak area is fully accounted for

76: 98% ee, not 97%

77: 97% ee, not 96%

78: 89% ee, not 90%.

83: 89% ee, not 90%.

85: 93% ee, not 90%.

86: 93% ee, not 94%.

Fig 4. a. You should explain in figure what it means to have %ee (%ee). What refers to what. Wouldn't this reaction be a sequential 7CR & 8CR not formal 6CR & 7 CR.

88: 98% ee, not 99%.

89: Should zoom in more to see proper curve and if it is measured correctly. When comparing to racemic version, the 2nd peak goes on for much longer.

90: Same as above.

Fig. 5, b.: All these figures need to be re-added, but higher quality. They are blurry.

Reviewer #3 (Remarks to the Author):

Response to Reviewer(s)' Comments

Reviewer #1 (Remarks to the Author):

Multicomponent reactions are extremely important in organic synthesis because they provide a complex structural unit by stitching readily available molecules together. Certainly, there has been a great deal of success in achieving such a reaction in the literature. However, the enantioselective version of those process are not so common especially when it comes to four component reactions. As mentioned by the authors the first successful example includes the enantioselective Ugi-4CR utilizing an aldehyde, an amine, a carboxylic acid and an isocyanide. This is followed by the reports from Feng and Hoveyda (Ref 21 and 22). In this report, the authors reported the first example enantioselective of four-component reaction of alcohols, diazoesters, enamines/indoles and aldehydes. The reaction affords products in good yields with good to excellent chemo-, diastereo- and enantioselectivity. The reaction seems to be very general and this referee has no doubt about its potential in terms of creating chiral building block and developing analogues multicomponent reactions of high magnitude. This referee is in favour supporting this work for publication in this journal after addressing the following comments.

1) The reports lack a variety of reviews on multi-catalysis where more than one catalysts work in tandem in relay or cooperative fashions. Just to quote a few: *Tetrahedron*, 2013, 69, 7171; *OBC*, 2012, 10, 211, *Chem. Rev.*, 2005, 105, 1001; *Coord. Chem. Rev.*, 2004, 248, 2365; *OBC*, 2015, 13, 8116; *Chem. Asian J.* 2010, 5, 422, *ACS Catal.* 2021, 11, 3891 etc.

Response: Thank you very much for your comments and suggestions. These all reviews mentioned above have been cited in the revised manuscript as follow: We have added seven references listed by reviewers in the manuscript as follow.

- 39 Fogg, D. E., & dos Santos, E. N. Tandem catalysis: a taxonomy and illustrative review. *Coord. Chem. Rev.* **248**, 2365-2379 (2004).
- 40 Wasilke, J. C., Obrey, S. J., Baker, R. T., & Bazan, G. C. Concurrent tandem catalysis. *Chem. Rev.* **105**, 1001-1020 (2005).
- 41 Zhou, J. Recent advances in multicatalyst promoted asymmetric tandem reactions. *Chem. Asian. J.* **5**, 422-434 (2010).
- 42 Patil, N. T., Shinde, V. S., & Gajula, B. A one-pot catalysis: the strategic classification with some recent examples. *Org. Biomol. Chem.* **10**, 211-224 (2012).
- 43 Pellissier, H., & Pellissier, H. Recent developments in enantioselective multicatalysed tandem reactions. *Tetrahedron* **69**, 7171-7210 (2013).
- 44 Inamdar, S. M., Shinde, V. S., & Patil, N. T. Enantioselective cooperative catalysis. *Org. Biomol. Chem.* **13**, 8116-8162 (2015).

45 Martinez, S., Veth, L., Lainer, B., & Dydio, P. Challenges and opportunities in multicatalysis. *ACS. Catal.* **11**, 3891-3915 (2021).

2) The control experiments are missing: 1) the reaction of enol (or its keto equivalent) with aldehyde and enamines/indoles 2) the reaction of alcohol and diazo compound with enamines/indoles equivalent (for example, appropriately substituted indole-3-carbinol)

Response: Thank you very much for your comments and suggestions. We have added the control experiment of keto with aldehyde and enamines/indoles as follow, which showed in Supplementary chapter 7.11.

Procedure

Firstly, we synthesized **ketone** (insertion products) by using diazo compounds and (4-bromophenyl)methanol according to the known reference. Secondly, we conducted the control reactions of ketone with aldehyde and enamines/indoles under the standard condition A/B. Thirdly, LC-MS was used to detect the existing products in the reaction. **Result:** Neither 4CR products were not detected by LC-MS.

NOTE: N.P is short for no product.

Response: Thank you very much for your comments and suggestions. We have added the control experiment of alcohol and diazo compound with enamines/indoles equivalent (for example, appropriately substituted indole-3-carbinol) as follow, which were presented in Supplementary chapter 7.12.

Procedure

Firstly, we synthesized **indole-3-carbinol derivative 6** by using 2-methyl-1H-indole-3-carbaldehyde and phenylacetylene according to the known reference. Secondly, we conducted the control reaction of alcohol and diazo compound with **6** under the standard condition A. Thirdly, LC-MS was used to detect the existing products in the reaction. **Result:** Target product **7** was not detected by LC-MS.

These four by-products could be detected by LC-MS.

NOTE: N.P is short for no product.

Procedure

Firstly, we synthesized compound **7** by using acetophenone and 4-nitrobenzaldehyde according to the known reference. Secondly, we conducted the control reaction of alcohol and diazo compound with by-product **7** under the standard condition B. Thirdly, LC-MS was used to detect the existing

products in the reaction. **Result:** Target product **83** was not detected by LC-MS.

These three by-products could be detected by LC-MS.

Reviewer #2 (Remarks to the Author):

This manuscript by Yu and coworkers reports an efficient enantioselective four-component reaction via in-situ two intermediates. The reaction proceeds with four-component that involve alcohols, diazoesters, enamines/indoles, and aldehydes and furnishes the corresponding coupling products in good yield under mild reaction conditions and shows broad substrate scope. Besides evaluating the synthetic scope of the reaction and the synthetic applications, they also elucidate its mechanism involving DFT computational work that was conducted to support the stereoselectivity of the Michael addition. However, the computations currently of the transition-metal catalysis are discussed in such a way as to suggest that these calculations support the current experimental results, which is currently not the case. Therefore, I either recommend the authors do the additional calculations in order to support the experimental data (such as a larger database method) or rewrite the discussion of the current DFT calculations. Specific comments and questions are provided below:

1. Though the authors propose the imine was involved in the Pd catalysis to close the catalysis cycle, the current calculations suggest that INT-11 to INT-7 (Fig.5a) is unlikely to happen smoothly, as the intermediates' barrier of 21.2 kcal/mol will not result in good yield at -10 °C. Additional calculations on the concerted pathway are needed to support the current experimental evidence.

Response: We appreciate the reviewer for the comments. Assuming the Pd-catalyzed decomposition of diazo ester as a first order reaction, the rate constant can be obtained as $k = 4.26 \times 10^{-4} \text{ s}^{-1}$ when the conversion was 99% at -10 °C after 3 h. According to Eyring equation, the corresponding reaction barrier was $\Delta G_{\text{exp}}^{\ddagger} = 19.4 \text{ kcal/mol}$, which was very close to our computed barrier ($\Delta G_{\text{calc}}^{\ddagger} = 21.0 \text{ kcal/mol}$), considering the 2-3 kcal/mol error for DFT calculations.

2. The authors mention “the substrate scope and limitation of this four-component reaction were then investigated (Fig.2).” on page 3, line 99, but there is no limitation result in Fig.2 or others. Please include some limitation results in the manuscript, it will be better for the chemists to study the research. For example, secondary alcohols, protected indoles, phenyl group instead of the methyl group at 2- position of indole, or N-heteroaryl group, etc.

Response: Thank you very much for your comments and suggestions. We have included some limitation results in the revised manuscript as follow, which have been highlighted in yellow color. Additionally, we investigated the substrate limitation of this A4CR. For example, secondary alcohol, tertiary alcohol, unprotected indoles, indole with a phenyl group at 2-position, and simple aryl aldehyde could not convert to their related products under the catalytic condition A (more detail see Supplementary Fig.S1b).

The yield of product was detected by LC-MS

Additionally, we investigated the substrate limitation of this A4CR. For example, secondary alcohol, tertiary alcohol, diazoacetate, alkyl enamine and simple aryl aldehyde could not convert to their related products under the catalytic condition B (more detail see Supplementary Fig.S2b).

The yield of product was detected by LC-MS

3. The authors should write the conditions in the footnote of Fig.2 and 3, such as the mmol of each substrate and the syringe addition time, etc.

Response: Thank you very much for your comments and suggestions.

We have added the detailed reaction conditions in the footnote of Fig.2 as follow, which have been

highlighted in yellow color in the revised manuscript.

Standard conditions: **1/2/3/4/5a/[Pd]/6a** = 0.6/0.3/0.2/0.6/0.1/0.01/0.02 mmol. **2, 3** in 4.0 mL dry DCE was added into a solution of **1, 4, 5a** (50 mol%), [Pd] (5 mol%), **6a** (10 mol%), and 4 Å MS (50 mg) in 4.0 mL dry DCE via a syringe pump for 60 mins, and the resulting mixture was stirred for another 12 hours at -20 °C.

We have added the detailed reaction conditions in the footnote of Fig.3 as follow, which have been highlighted in yellow color in the revised manuscript.

Standard condition: **1/2/3/4/5b/[PdCl(allyl)]₂/6b** = 0.18/0.18/0.18/0.15/0.075/0.0075/0.015 mmol, **2** and **3a** in 1.0 mL dry DCM were added into a solution of **1, 4, [PdCl(allyl)]₂, 6b**, aniline **5b** and 100 mg 4 Å MS in 1.5 mL dry DCM via a syringe pump under a nitrogen atmosphere for 3 hours, and the resulting mixture was stirred for another 1 hour at -10 °C.

4. In Fig.2, the structure of products 22 to 27, the authors should use R2 instead of R1 and use R3 instead of R2 to be consistent with the general structure of substrate 3 on the top of Fig.2.

Response: Thank you very much for your comments and suggestions. We have used R² instead of R¹ and used R³ instead of R² to be consistent with the general structure of substrate 3. The revised structures were as follows with a red rim.

5. It's better to involve the TS barrier in the proposed reaction mechanism (Fig.5a).

Response: Thanks for the reviewer's suggestion. In Fig.5a, we have presented the key transition states and intermediates along with Gibbs free energies under the corresponding structures. In the main text, we have given the reaction barriers of the formation of the iminium ($\Delta G^\ddagger = 19.4$ kcal/mol) and enol ($\Delta G^\ddagger = 21.0$ kcal/mol) intermediates, respectively. Moreover, detailed free energy profiles of the four-component reaction have been presented in the Figure S5-S7.

6. I would like to suggest the authors use a larger database method to calculate the stereoselectivity including a Boltzmann calculation in further research.

Response: Thanks for the reviewer's kind suggestion. During the optimizations of involved transition states and intermediates, extensive conformational searches were conducted and only the lowest energy conformers were shown in this work. For example, a series of different conformers of the stereocontrolling transition states were obtained as TS-SR, TS-SR-a, ..., TS-SR-i (see below), but only the transition state TS-SR, the structure with the lowest energy, was selected as optimal conformer in our study. The considerable energy difference (> 1 kcal/mol) between TS-SR and other conformers indicates TS-SR played a predominate role in forming the desired product. As a result, only TS-SR was considered when calculating the stereoselectivity. The abovementioned results have been added in the revised Supplementary Information as Figure S10.

Figure S10. The different conformers of the stereocontrolling transition states **TS-SR** for the asymmetric four-component reaction using CPA **6b** and diazoester **2a**. The Gibbs free energies were computed with CPCM(DCM)-M06-2X/6-311+G(d,p)//CPCM(DCM)-B3LYP/6-31G(d).

Reviewer #3 (Remarks to the Author):

1.) The noteworthy results of this paper is an example of a highly enantioselective four-component reaction with a large reaction scope.

Response: We appreciate the reviewer for the positive and kind comments.

2.) Established literature for enantioselective reactions with diazoesters do exist, but are typically only two-three component reactions. This being a four component reaction with high ee and dr as well as a DFT-mechanistic study and large scope add to its value. Additionally, having a racemic product to compare to the enantioselective product further validated these claims. I would recommend to have a figure introducing to the readers previous work of enantioselective three/four component reactions involving the use of CPA/diazo esters, see: *Molecules* 2021, 26, 6563

Response: Thank you very much for your comments and suggestions. Our research group has developed an array of asymmetric multi-component reactions (AMCRs) via electrophilic trapping of ylides or zwitterions generated from metal carbenes, which are realized under the cooperative catalysis of two catalysts, including metal catalyst and chiral phosphoric acid (CPA). To let readers know more clearly about the multi-component reactions involving the use of CPA/diazo esters developed by our research group. According to reviewer's suggestion, we add a figure containing some representative examples of three/four component reactions involving the use of CPA/diazo esters to introduce our previous works as follows and cited some references (ref 25-30) about our previous works. Moreover, the important review on multicomponent reactions (*Molecules* 2021, 26, 6563) has been cited as ref 22 in the revised manuscript.

a. Lego-like synergistic organic synthesis

b. General process of trapping type 3/4CRs

c. The representative examples of 3/4CRs involving the use of CPA/diazo esters

1. 3CRs involving the oxonium ylide (J. Am. Chem. Soc. 2008, 130, 7782-7783)

2. 3CRs involving the ammonium ylide and the use of CPA/diazo esters (Nat. Commun. 2015, 6, 5801)

3. 3CRs involving the zwitterion and the use of CPA/diazo esters (Nat. Chem. 2012, 4, 733-738)

4. 4CRs involving the oxonium ylide and the use of CPA/diazo esters (Chem. Commun., 2008, 6564-6566)
(Electrophiles was imine formed by amine and aldehyde in situ)

d. This 4CR via intermediate cross interception

Fig.1 | Patterns of chemical reactions. **a**, Lego-like synergistic organic synthesis. **b**, General process of trapping type MCRs developed by our group. **c**, The representative examples of 3/4CRs involving the use of CPA/diazo esters. **d**, This 4CR via intermediate cross interception. **NOTE:** Nu is short for nucleophile, Co-cat. Is short for co-catalyst, CPA is short for chiral phosphoric acid, [ML_n] is short for metal catalyst, [Am] is short for amine catalyst, [H⁺] is short for Brønsted acid catalyst.

New added references in the manuscript:

- 22 Bakulina, O., Inyutina, A., Dar'in, D., & Krasavin, M. Multicomponent reactions involving diazo reagents: a 5-year ppdate. *Molecules* **26** (2021).
- 25 Kang, Z. *et al.* Ternary catalysis enabled three-component asymmetric allylic alkylation as a concise track to chiral alpha,alpha-disubstituted ketones. *J. Am. Chem. Soc.* **143**, 20818-20827 (2021).
- 26 Hu, W. *et al.* Cooperative catalysis with chiral Brønsted acid-Rh₂(OAc)₄: highly enantioselective three-component reactions of diazo compounds with alcohols and imines. *J. Am. Chem. Soc.* **130**, 7782-7783 (2008).
- 27 Qiu, H. *et al.* Highly enantioselective trapping of zwitterionic intermediates by imines. *Nat. Chem.*, **4** 733-738 (2012).
- 28 Zhang, D., Zhou, J., Xia, F., Kang, Z. & Hu, W. Bond cleavage, fragment modification and reassembly in enantioselective three-component reactions. *Nat. Commun.* **6**, 5801 (2015).
- 29 Xu, X., Zhou, J., Yang, L., Hu, W. Selectivity control in enantioselective four-component reactions of aryl diazoacetates with alcohols, aldehydes and amines: an efficient approach to synthesizing chiral beta-amino-alpha-hydroxyesters. *Chem. Commun.*, 6564-6566 (2008).
- 30 Guo, X., & Hu, W. Novel multicomponent reactions via trapping of protic onium ylides with electrophiles. *Acc. Chem. Res.* **46**, 2427-2440 (2013).

3.) Based on chiral HPLC/NMR and previous work involving the use of these catalysts, and DFT calculations, the results of this work are sufficiently supported.

3a) Analysis of the %ee via the chiral HPLC chromatogram needs to be relooked at. When comparing the racemic to the chiral reaction, the amount measured for the minor peak is consistently smaller when compared to the racemic one. Therefore, a higher ee% is being measured that what is obtained. The chiral HPLC chromatogram needs to be zoomed in further and the area measured needs to account for the entire curve.

Response: Thank you very much for your comments and suggestions, we really appreciated your thoroughness and attention to detail. We answer your questions in the attachment by point to point, which have been highlighted in yellow color in the revised manuscript/supplementary information.

(1) Figure 1, b: The arrow going from the enol to the iminium is not correct. During nucleophilic attack onto an electrophile, you draw the arrow going from the bond to an atom. Draw the full arrow

cascade, arrow from OH going into bond to form ester, then arrow from the double bond to atom of iminium, then arrows are needed on iminium to show flow of electrons. You should denote what Am and MLn stands for. It may be obvious for some, but not others.

Response: Thank you very much for your kind suggestions. As you suggested, we patched up the missing arrows, which was updated in the Figure 1d in the revised manuscript.

d. This 4CR via intermediate cross interception

(2) Table S1. Yields should be reported as 42 not 42.0, don't report yields with a decimal point.

Response: Thank you very much for your comments and suggestions. The format of yields has been corrected according to reviewer (highlight in yellow color).

Table S1 Condition Optimization of Racemic Four-component Reaction ^a

Entry	III-5	x (equiv.)	y (equiv.)	z (equiv.)	w (equiv.)	dr ^b	Yield (%) ^c
1	5a	1.2	1.5	1.2	1.0	>20:1	25
2	S5b	1.2	1.5	1.2	1.0	-	ND
3	S5c	1.2	1.5	1.2	1.0	-	ND
4	S5d	1.2	1.5	1.2	1.0	-	ND
5	S5e	1.2	1.5	1.2	1.0	-	ND
6	5a	1.2	2.0	1.2	1.0	>20:1	20
7	5a	1.2	1.5	1.0	3.0	>20:1	44
8	5a	1.2	1.5	1.0	4.0	>20:1	42
9	5a	1.5	1.5	1.0	3.0	>20:1	42
10	5a	2.0	1.5	1.0	3.0	>20:1	54
11	5a	3.0	1.5	1.0	3.0	>20:1	63
12	5a	4.0	1.5	1.0	3.0	>20:1	56

(3) Figure 2, 12: 93% ee, not 94%.

Response: Thank you very much for your thoroughness and attention to detail. The ee value of **12** has been corrected from 94% to 93% in Figure 2 (highlight in blue as follow).

(4)15: 91% ee, not 92%.

Response: Thank you very much for your thoroughness and attention to detail. The ee value of **15** has been corrected from 92% to 91% in figure 2 (highlight in blue as follow).

(5)18: 92% ee

Response: Thank you very much for your thoroughness and attention to detail. The ee value of **18** has been corrected to 92% in figure 2 (highlight in blue as follow).

(6)21: Crystal structure should have higher quality, when zooming in it is really blurry.

Response: Thank you very much for your comments and suggestions. As you suggested, the picture of crystal structure of 21 has been replaced with a higher quality, which was updated in the Figure 2.

(7)22: 95% ee, not 96%.

Response: Thank you very much for your thoroughness and attention to detail. The ee value of **22** has been corrected from 96% to 95% in figure 2 (highlight in blue as follow).

(8)23: 87% ee, not 88%.

Response: Thank you very much for your thoroughness and attention to detail. The ee value of **23** has been corrected from 88% to 87% in figure 2 (highlight in blue as follow).

(9)24: Yield states 89% in SI, but 85% in paper, which is it?

Response: Thank you very much for your thoroughness and attention to detail. The yield value of **24** has been corrected from 85% to 89% in figure 2 (highlight in blue as follow).

(10)25: 92% ee, not 93%.

Response: Thank you very much for your thoroughness and attention to detail. The ee value of **25** has been corrected from 93% to 92% in figure 2 (highlight in blue as follow).

(11)27: 93% ee, not 94%.

Response: Thank you very much for your thoroughness and attention to detail. The ee value of **27** has been corrected from 94% to 93% in figure 2 (highlight in blue as follow).

(12)28: 97% ee. Not 98% ee. Also, the Chiral IA graph could be zoomed in a bit. It looks like the area for the peak at 28 minutes is not fully covered.

Response: Thank you very much for your thoroughness and attention to detail. The ee value of **28** has been corrected from 98% to 97% in figure 2 (highlight in blue as follow). Also, we zoomed HPLC chromatogram of chiral compound locally, the larger version of HPLC chromatogram of chiral **28** was added in the revised Supplementary information.

The HPLC chromatogram of chiral 28

The larger version of HPLC chromatogram of chiral 28

(13)29: 86% ee, not 87%.

Response: Thank you very much for your thoroughness and attention to detail. The ee value of **29** has been corrected from 87% to 86% in figure 2 (highlight in blue as follow).

(14)34: SI states 68% yield, but paper states 77% yield, which is it?

Response: Thank you very much for your thoroughness and attention to detail. The yield value of **34** has been corrected from 77% to 68% in figure 2 (highlight in blue as follow).

(15)36: 95% ee, not 96% ee.

Response: Thank you very much for your thoroughness and attention to detail. The ee value of **36** has been corrected from 96% to 95% in figure 2 (highlight in blue as follow).

(16)37: 94% ee, not 95% ee.

Response: Thank you very much for your thoroughness and attention to detail. The ee value of **37** has been corrected from 95% to 94% in figure 2 (highlight in blue as follow).

(17)39: 91% ee, not 92% ee.

Response: Thank you very much for your thoroughness and attention to detail. The ee value of **39** has been corrected from 92% to 91% in figure 3 (highlight in blue as follow).

(18)40: 96% ee, not 98% ee.

Response: Thank you very much for your thoroughness and attention to detail. The ee value of **40** has been corrected from 98% to 96% in figure 3 (highlight in blue as follow).

(19)43-45: Chiral IA graph should be zoomed in more; it might not make a difference but markers for minor product don't look them cover the full peak area.

Response: Thank you very much for your comments and suggestions. We zoomed HPLC chromatogram of chiral compound locally, the larger version of HPLC chromatogram of chiral **43-45** were added in the file.

HPLC chromatogram of chiral 43

The larger version of HPLC chromatogram of chiral 43

HPLC chromatogram of chiral 44

The larger version of HPLC chromatogram of chiral 44

HPLC chromatogram of chiral 45

The larger version of HPLC chromatogram of chiral 45

(20)46: 93% ee, not 94%.

Response: Thank you very much for your thoroughness and attention to detail. The ee value of **46** has been corrected from 94% to 93% in figure 3 (highlight in blue as follow).

(21)48: 97% ee, not 98%.

Response: Thank you very much for your thoroughness and attention to detail. The ee value of **48** has been corrected from 98% to 97% in figure 3 (highlight in blue as follow).

(22)49: 97% ee, not 98%. Also, area under curve is not measured correctly for minor peak, need to zoom in more and measure correct area.

Response: Thank you very much for your thoroughness and attention to detail. The ee value of **49** has been corrected from 98% to 97% in figure 2 (highlight in blue as follow). Also, we zoomed HPLC chromatogram of chiral compound locally, the larger version of HPLC chromatogram of chiral **49** was added in the file.

HPLC chromatogram of chiral 49

The larger version of HPLC chromatogram of chiral 49

(23)51: 93% ee, not 94%.

Response: Thank you very much for your thoroughness and attention to detail. The ee value of **51** has been corrected from 94% to 93% in figure 3 (highlight in blue as follow).

(24)53: SI states 55% yield and paper states 60% yield, which is it. Paper states 93% ee, SI states 96% ee. Based on graph, 93% ee

Response: Thank you very much for your thoroughness and attention to detail. The yield value of **53** has been corrected from 60% to 55% in figure 3, and the ee value was 93% (highlight in blue as follow).

(25)54: 92% ee, not 93% ee.

Response: Thank you very much for your thoroughness and attention to detail. The ee value of **54** has been corrected from 93% to 92% in figure 3 (highlight in blue as follow).

(26)55: 89% ee, not 90% ee.

Response: Thank you very much for your thoroughness and attention to detail. The ee value of **55** has been corrected from 90% to 89% in figure 3 (highlight in blue as follow).

(27)58: 91% ee. not 90%.

Response: Thank you very much for your thoroughness and attention to detail. The ee value of **51** has been corrected from 90% to 91% in figure 3 (highlight in blue as follow).

(28)59: Can just write 9:1, consistent with how you wrote it for previous molecules (20:1).

Response: Thank you very much for your thoroughness and attention to detail. The format of dr value of **59** has been corrected from 90:10 to 9:1 in figure 3 (highlight in blue as follow).

(29)65: 92% ee, not 90%.

Response: Thank you very much for your thoroughness and attention to detail. The ee value of **65** has been corrected from 90% to 92% in figure 3 (highlight in blue as follow).

(30)68: for ee so low the graph should be zoom in a little bit more.

Response: Thank you very much for your comments and suggestions. We zoomed HPLC chromatogram of chiral compound locally, the larger version of HPLC chromatogram of chiral **68** was added in the file.

HPLC chromatogram of chiral 68

The larger version of HPLC chromatogram of chiral 68

(31)72: Based on racemic peak, peak looks like it is supposed to be very wide and the peak area that you

highlighted does not match how it would match in racemic peak. Again, this should be zoomed in greater so that the correct area can be visualized better.

Response: Thank you very much for your comments and suggestions. We zoomed HPLC chromatogram of chiral compound locally, the larger version of HPLC chromatogram of chiral **72** was added in the file.

HPLC chromatogram of chiral 72

The larger version of HPLC chromatogram of chiral 72

(32)73: Same issue as above.

Response: Thank you very much for your comments and suggestions. We zoomed HPLC chromatogram of chiral compound locally, the larger version of HPLC chromatogram of chiral **73** was added in the file.

HPLC chromatogram of chiral 73

The larger version of HPLC chromatogram of chiral 73

(33)74: Same issue as above, doesn't look like peak area is fully accounted for.

Response: Thank you very much for your comments and suggestions. We zoomed HPLC chromatogram of chiral compound locally, the larger version of HPLC chromatogram of chiral **74** was added in the file.

HPLC chromatogram of chiral 74

The larger version of HPLC chromatogram of chiral 74

(34)76: 98% ee, not 97%

Response: Thank you very much for your thoroughness and attention to detail. The ee value of **76** has been corrected from 97% to 98% in figure 3 (highlight in blue as follow).

(35)77: 97% ee, not 96%

Response: Thank you very much for your thoroughness and attention to detail. The ee value of **77** has been corrected from 96% to 97% in figure 3 (highlight in blue as follow).

(36)78: 89% ee, not 90%.

Response: Thank you very much for your thoroughness and attention to detail. The ee value of **78** has been corrected from 90% to 89% in figure 3 (highlight in blue as follow).

(37)83: 89% ee, not 90%.

Response: Thank you very much for your thoroughness and attention to detail. The ee value of **83**

has been corrected from 90% to 89% in figure 3 (highlight in blue as follow).

(38)85: 93% ee, not 90%.

Response: Thank you very much for your thoroughness and attention to detail. The ee value of **85** has been corrected from 90% to 93% in figure 3 (highlight in blue as follow).

(39)86: 93% ee, not 94%.

Response: Thank you very much for your thoroughness and attention to detail. The ee value of **85** has been corrected from 94% to 93% in figure 3 (highlight in blue as follow).

(40)Fig 4. a. You should explain in figure what it means to have %ee (%ee). What refers to what. Wouldn't this reaction be a sequential 7CR & 8CR not formal 6CR & 7 CR.

Response: Thank you very much for your comments and suggestions. In order to avoid feeling confused, we corrected the content of the Fig. 4a (with a red rim) and added some notes in the footnote of Fig.4

Fig.4a | Formal higher-order MCRs design. **NOTE:** The other diastereomer **87'** of compound **87** has the same ee value (92% ee); The other diastereomer **88'** of compound **88** has the same ee value (98% ee); The other diastereomer **91'** of compound **91** has the same ee value (98% ee).

(41)88: 98% ee, not 99%.

Response: Thank you very much for your thoroughness and attention to detail. The ee value of **85** has been corrected from 99% to 98% in figure 3 (highlight in blue as follow).

(42)89: Should zoom in more to see proper curve and if it is measured correctly. When comparing to racemic version, the 2nd peak goes on for much longer.

Response: Thank you very much for your comments and suggestions. We zoomed HPLC

chromatogram of chiral compound locally, the larger version of HPLC chromatogram of chiral **89** was added in the file.

HPLC chromatogram of chiral **89**

The larger version of HPLC chromatogram of chiral **89**

(43)90: Same as above.

Response: Thank you very much for your comments and suggestions. We zoomed HPLC chromatogram of chiral compound locally, the larger version of HPLC chromatogram of chiral **90** was added in the file.

HPLC chromatogram of chiral **90**

The larger version of HPLC chromatogram of chiral 90

(44) Fig. 5, b.: All these figures need to be re-added, but higher quality. They are blurry.

Response: Thanks for your suggestion. The original figures in Fig. 5b have been replaced by corresponding ones with higher quality (see below) in the revised manuscript.

b Transition states of asymmetric Michael addition and origin of stereoselectivity

Below are updated Figure 2, Figure 3, and Figure 4.

Fig.2 | Substrate scope of enantioselective four-component reactions of alcohol, diazoester, indole and aldehyde derivatives. Standard conditions: **1/2/3/4/5a/[Pd]/6a** = 0.6/0.3/0.2/0.6/0.1/0.01/0.02 mmol. **2, 3** in 4.0 mL dry DCE was added into a solution of **1, 4, 5a** (50 mol%), [Pd] (5 mol%), **6a** (10 mol%), and 4 Å MS (50 mg) in 4.0 mL dry DCE via a syringe pump for 60 minutes, and the resulting mixture was stirred for another 12 hours at -20°C .

Fig.3 | Substrate scope of enantioselective four-component reactions of alcohol, diazoester, enamine and aldehyde derivatives. Standard condition: **1/2/3/4/5b/[PdCl(allyl)]₂/6b** = 0.18/0.18/0.18/0.15/0.075/0.0075/0.015 mmol, **2** and **3a** in 1.0 mL dry DCM were added into a solution of **1**, **4**, [PdCl(allyl)]₂, **6b**, aniline **5b** and 100 mg 4 Å MS in 1.5 mL dry DCM via a syringe pump under a nitrogen atmosphere for 3 hours, and the resulting mixture was stirred for another 1 hour -10 °C.

Fig.4 | Formal higher-order MCRs design and synthetic applications. a, Higher-order MCRs design. b, Synthetic applications. NOTE: The other diastereomer **87'** of compound **87** has the same ee value (92% ee); The other diastereomer **88'** of compound **88** has the same ee value (98% ee); The other diastereomer **91'** of compound **91** has the same ee value (98% ee).

3b) Additionally, the grammar/sentence structure needs to be revised for the whole paper. Such recommendations were given up to line 49.

Response: Thanks very much for helping us polish our manuscript. Before resubmission,-we have sent the manuscript to the English language editing service and made careful polish we also invited

a native speaker for help in English editing. Those contents have been highlighted in yellow color.

The methodology is sound and meets the expected standards. Given the large reaction scope, optimization table, and previous literature, it is expected that this work would be reproducible.

Reviewer,

Kevin Schofield

REVIEWERS' COMMENTS

Reviewer #1 (Remarks to the Author):

The manuscript has been revised in the comments of the referee. Hence, the manuscript is now suitable for publication.

Reviewer #2 (Remarks to the Author):

I think that the revised manuscript and SI are great and suitable for publication in Nature Communications, especially considering the further DFT study and the substrate limitation study. This work presents over 100 examples of a four-component reaction with excellent stereoselective, it is an important contribution to the study of the multicomponent reaction.

Reviewer #3 (Remarks to the Author):

All my primary concerns with the first draft of the manuscript have been addressed and I believe this paper is ready for publication.